# Exploring exercise, emotional eating, and body dissatisfaction in depression risk: Using structural equation modeling

**Zhimin Yi, Mingtao Chen◉, Renzhao Huang◉, Weiguo Chen◉, Huigen Liu◉, Guoqiu Liu◉, Ming Hao◉***

School of Public Health and Health Management, Gannan Medical University, Ganzhou City, Jiangxi Province, China

◉ These authors contributed equally to this work.
* hm48922200@yahoo.co.jp

## Abstract

In order to fill the relevant research gap on the influence of body dissatisfaction on depression in Chinese college students, we used structural equation modeling to explore the relationships between physical activity, body dissatisfaction, eating behavior, and depression. Participants were 1,714 undergraduate students in Southern China. Exercise served as a significant mediator between body mass index and depression, as well as between body dissatisfaction and emotional eating, accounting for 30% and 19.8% of the total effects, respectively. Emotional eating behavior mediated the relationship between exercise and depression. This study underscores the influences of body dissatisfaction on both physical inactivity and emotional eating patterns among Chinese university students. It suggests that reducing body dissatisfaction and emotional eating holds significant potential for preventing depression.

## 1. Introduction

Depression is a leading cause of mental disorders worldwide [1], increasingly affecting younger populations [2]. It is estimated that depression is the third leading cause of disability globally [3]. A US study revealed that from 2015 to 2020, depression was concentrated among individuals aged 18–25 and 12–17 [4]. A study of college students in southern China reported a weighted depression rate of 14.7% (95% CI: 14.0%-15.5%) using the self- rated depression scale [5]. Given the high prevalence of depression among adolescents and the significant public health burden it poses, there is an urgent need to focus on the psychological status of college students and conduct systematic investigations. Psychological challenge pose an even greater burden for college students who lack resilience and coping skills [6]. Depression is a high-risk factor for suicide among adolescents, seriously affecting their physical and mental health, and deserves our utmost attention [7]. Many factors influence

**Data availability statement:** All relevant data are within the manuscript and its Supporting Information files.

**Funding:** This study was supported by the Starting Research Fund from the Gannan Medical University (QD202121), Humanities and Social Sciences Fund from the Ministry of Education of China (22YJC630085), Humanities and Social Sciences of Jiangxi University in 2023 (GL23114). There was no additional external funding received for this study.

**Competing interests:** The authors have declared that no competing interests exist.

depression among adolescents [8]. Especially, there have been many studies on the relationship between depression and people's exercise and eating behaviors [9,10]. However, there are few studies on the combined effects of exercise and dietary behaviors on depression. For example, a recent systematic review highlighted that while individual studies have examined the links between diet and depression or exercise and depression, the combined effects of these lifestyle factors are less understood [9]. Another study emphasized the need for more research on the integrated impact of physical activity and diet on mental health outcomes [10]. Considering the significant impact of lifestyle on mental health among college students, this study will investigate the combined effect of exercise and dietary behaviors on depression.

There are no definitive preventive or therapeutic measures for depression. The relationship between exercise and depression has garnered significant interest [11]. A representative cross-sectional study in China shows that nearly a quarter of adolescents do not engage in physical activity for almost a full day and overall physical activity levels are decreasing year by year [12]. Owing to the COVID-19 pandemic, many countries implemented social distancing or isolation protocols, leading to decrease physical activity levels and increased sedentary behaviors [13]. Research indicates that exercise significantly impacts the severity and relief of depression [14,15]. For example, a meta-analysis demonstrated that exercise has a moderate antidepressant effect, reducing symptoms of depression across various populations [15]. Similarly, a recent systematic review highlighted a negative dose-response relationship between physical activity levels and depression risk, further supporting the role of exercise in depression management [14]. It is considered a low-cost and easy-to-implement therapeutic option that can be used as monotherapy, adjunctive therapy, or in combination with other therapies to alleviate depression [16]. Physical activity is often safer and more accessible than other clinical depression treatments.

The relationship between diet and mental health has been a long-standing topic of interest [17]. In 2022, according to a study, eating disorders have risen by 80% over the past five years [18]. Dietary changes may affect mental illness through direct effects on mood, and the development of mental illness may lead to changes in eating habits [19]. Poor emotional states may lead to unhealthy eating habits related to negative emotions [20]. Emotional eating behavior refers to how emotional factors influence eating habits, particularly how negative emotions can lead individuals to eat to alleviate those emotions [21]. During college years, when students do not receive parental supervision, eating behaviors change dramatically, and unhealthy eating behaviors becomes increasingly common, which directly affects students' emotional state [22]. Therefore, it is important to explore the relationship between emotional eating and depression.

Many studies have examined the direct effects of exercise and dietary behaviors on depression, however, this study adds body dissatisfaction as a variable to explore their interactive effects. With entering the university, students pay more attention to their own body image [23]. Body dissatisfaction refers to the negative feelings that individuals experience due to inconsistencies between their actual and

desired appearance or weight [24]. Body dissatisfaction is a negative attitude towards one's own body due to perceptions, thoughts, and feelings about one's body image [25]. Body weight dissatisfaction represents a specific dimension of body dissatisfaction, characterized by a discrepancy between actual and ideal BMI, particularly among individuals with obesity [26]. College students' body dissatisfaction is closely related to eating behaviors and exercise [22,27]. As students enter college, they gain more freedom and experience a changing social environment, which can prompt dramatic lifestyles changes [22]. In addition to alterations in physical activity and eating behaviors, the participants also exhibited romantic needs and expressed a desire to seek a significant other. Therefore, including body dissatisfaction as a variable is particularly important for our study.

Numerous previous studies have analyzed factors contributing to depression using linear regression or logistic regression, often neglecting the indirect effects of variables on one another [28,29]. Studying the interrelations of several difficult-to-measure variables presents a challenge, and disregarding these issues can yield inaccurate results. Utilizing structural equation modeling can help address these challenges [30]. Therefore, we aimed to explore the relationship between exercise, body dissatisfaction, dietary behavior, and depression using structural equation modeling, and attempted to find ways to alleviate depression. The result of this study can aid in promoting sport and diet among Chinese college students to help them gain a clear understanding about depression with hopes of attaining good physical and mental health in the future.

In summary, depression represents a significant mental health issue among college students. Elucidating the factors that contribute to depression is essential for devising effective interventions. Prior research has established physical activity, emotional eating, and body dissatisfaction as key factors influencing mental health. However, the interrelationships among these factors and their collective impact on depression remain to be fully explored. Thus, the present study aims to examine the combined effects of physical activity, emotional eating, and body dissatisfaction on depression risk among Chinese college students using structural equation modeling.

## 2. Materials and methods

### 2.1. Participants

This study was conducted in accordance with the Helsinki Declaration. This study has been approved by the Ethics Committee of Gannan Medical University (NO.2021110) during the investigation process. In accordance with these policies, investigators provided research participants with detailed explanations about the data being collected in this study and confirmed that it is being used only for scientific research purposes and not being disclosed to the public. All participants provided written informed consent.

This study employed structural equation modeling (SEM), which focused on eating behavior, with 33 observed variables, and Self-Depression Scale (SDS) scale, with 20 observed variables. Therefore, we estimated the number of parameters using the N: q rule, with a sample size ratio of 10:1 for the SEM [31]. The minimum sample size required was 530. The final sample size included in our statistical analysis was 1714, which satisfied the sample size required for SEM.

An expansive college in Southern China was selected for this study, and standardized training was provided to those who completed the survey before the study began. Recruitment of participants was conducted through oral publicity and distribution of pamphlets in study rooms.

This cross-sectional study selected a comprehensive university in southern China and recruited 1,900 college students between 17 September and 23 November 2023 as study participants. Random sampling was used select students attending evening study halls. All participants were asked to complete anthropometric measurements and three questionnaires: Physical Activity Rating Scale-3 (PARS-3), Chinese version of the Dutch Dietary Behavior Questionnaire (C-DEBQ), the Self-Depression Scale (SDS), and a body dissatisfaction questionnaire. Students over the age of 24 were excluded from the study due to their low representation. A total of 1,714 students (aged 18–24 years; men:933; women:781) with complete data were included in the analysis.

## 2.2. Measurements

This study employed the widely used German height scale (Seca 213, Germany, accuracy of 0.1 cm) and a Japanese weight-measuring device (accuracy of 0.1 kg). Body mass index (BMI, kg/m$^2$) was calculated based on height and weight measurements, which were recorded by experienced personnel.

The questionnaire method was employed to investigate the ideal weight of college students. Ideal BMI was calculated based on actual height and ideal weight, and body dissatisfaction was calculated by combining the actual and ideal BMI values. The difference between the actual and ideal BMI values was used as the body weight dissatisfaction score, measuring body dissatisfaction among Chinese college students [26].

## 2.3. Questionnaire content

The questionnaire collected students' basic information, including age, muscle mass, gender, height, weight, body dissatisfaction index and monthly living expenses. Additionally, it comprised three scales to assess exercise levels, one to measure emotional states, and one to evaluate eating behaviors.

## 2.4. Criteria for division

### 2.4.1. SDS depression level classification.
The total raw score of the SDS comprised the sum of scores from the 20 items (raw score X). This raw score was then converted into a standard score (index score, Y), calculated as $Y = (1.25X)$. Items 1, 3, 4, 7, 8, 9, 10, 13, 15, and 19 were positively scored, with options A, B, C, and D assigned 1, 2, 3, and 4 points, respectively. Conversely, items 2, 5, 6, 11, 12, 14, 16, 17, 18, and 20 were negatively scored and assigned 4, 3, 2, and 1 point, respectively. Depression was judged based on the following criteria: not depressed: < 53 points, depressed: ≥ 53 points.

### 2.4.2. PARS-3.
The PARS-3 assesses physical activity by considering its intensity, time, and frequency. It calculates physical activity as the product of intensity, time, and frequency. Intensity levels are categorized as follows: 1. light exercise (e.g., walking, which does not cause changes in shortness of breath), 2. low-intensity but not stressful exercise (e.g., yoga, jumping rope), 3. moderate-intensity exercise (e.g., cycling), 4. exercise causing shortness of breath and sweating but not long-lasting (e.g., badminton, basketball, tennis), and 5. exercise causing significant shortness of breath, sweating, and lasting a long time (e.g., yoga, swimming). Participants reported the duration and frequency of each intensity level per session and per month. Intensity and frequency were graded on a scale of 1–5, with 1 indicating low intensity or frequency and 5 indicating high intensity or frequency. Time was graded from 1–5, with 0–4 points assigned, respectively. The criteria for classifying the amount of exercise were as follows: *low amount of exercise*: ≤ 19 points, *medium amount of exercise*: 20–42 points, and *high amount of exercise*: ≥ 43 points.

### 2.4.3. C-DEBQ.
The C-DEBQ comprises 33 questions divided into three dimensions assessing adolescents' emotional, external, and restrained eating styles. Emotional eating consists of 13 questions, such as "When you are angry, do you have a desire to eat?" External eating behaviors are measured using ten questions, such as "Do you eat more than usual when you see other people eating?" Restrictive Eating Behaviors are measured using ten questions, such as, "Do you find it difficult to resist tasty food?" Responses were rated on a five-point Likert scale (1 = never; 5 = always). The total score for each dimension is calculated by summing the scores of the respective items. The possible range of scores for each dimension is as follows: emotional eating (13–65), external eating (10–50), and restrained eating (10–50). Higher scores indicate a greater degree of the specific eating behavior.

## 2.5. Data collation

The collected data underwent consolidation and the removal of invalid questionnaires. Valid questionnaires were processed using EpiData software to ensure data consistency and accuracy.

## 2.6. Statistical methods

First, we examine the normality of the continuous variables. Sex-specific differences in age, muscle mass, body dissatisfaction, BMI, physical activity score, and three dietary behaviors, along with depressive symptoms, were analyzed using two independent sample nonparametric tests (two-tailed). Chi-square tests were employed to assess sex-specific differences in BMI, physical activity level, and depressive symptoms. Using Pearson correlation analysis to examine the degree of association between BMI, body dissatisfaction, physical activity score, emotional eating score, external eating score, and depression. The multiple analysis method in JMP16.1 was utilized for structural equation modeling, with BMI, body dissatisfaction, exercise, emotional eating, and depression selected as predictor or outcome variables. The relationship between variable pairs was examined, and the overall model fit was assessed. Additionally, the mediating effect of BMI on depression, exercise and depression, emotional eating, and body dissatisfaction was evaluated. Statistical significance was set at $p < 0.05$. All statistical analyses were conducted using SPSS 26.0 and JMP 16.1 software (SAS Institute Inc., Cary, NC, USA).

## 3. Results

Table 1 presents participants' basic characteristics, exercise habits, dietary behaviors, and depression status. Men exhibited a slightly higher average BMI (22.24 kg/m²) compared with women (21.23 kg/m²), with a significant difference in body dissatisfaction ($p < 0.001$). Women reported nearly twice the level of dissatisfaction with their body weight compared to the men (2.09 versus 0.97). Men demonstrated higher exercise scores and muscle mass than women ($p < 0.001$). Conversely, the women scored higher in all three dietary categories compared with men ($p < 0.001$). Additionally, women had higher

**Table 1. Gender differences in participants' basic characteristics, exercise and dietary behaviors.**

| | Mean±SD or n (%) | | | P |
|---|---|---|---|---|
| | Men (n=933) | Women (n=781) | Total (1714) | |
| Age | 19.79±2.02 | 19.25±1.66 | 19.55±1.88 | <.001 |
| Muscle mass (kg) | 51.07±7.72 | 36.33±4.55 | 44.36±9.78 | <.001 |
| Body dissatisfaction (kg/m²) | 0.97±3.24 | 2.09±2.27 | 1.48±2.89 | <.001 |
| BMI(kg/m²) | 22.24±3.68 | 21.23±3.03 | 21.78±3.44 | <.001 |
| BMI category | | | | |
| Underweight | 87 (9.3) | 83 (10.6) | 170 (9.9) | <.001 |
| Normal | 601 (64.4) | 588 (75.3) | 1189 (69.4) | |
| Overweight | 170 (18.2) | 79 (10.1) | 249 (14.5) | |
| Obesity | 75 (8) | 31 (4) | 106 (6.2) | |
| Physical activity score | 21.77±21.21 | 11.42±14.71 | 17.05±19.23 | <.001 |
| Activity level category | | | | |
| Low exercise | 563 (60.3) | 659 (84.4) | 1222 (71.3) | <.001 |
| Medium exercise | 203 (21.8) | 75 (9.6) | 278 (16.2) | |
| High exercise | 167 (17.9) | 47 (6) | 214 (12.5) | |
| Eating behaviour | | | | |
| Restrained eating score | 24.56±8.1 | 28.47±7.46 | 26.34±8.05 | <.001 |
| Emotional eating score | 23.78±9.96 | 26.7±10.02 | 25.11±10.09 | |
| External eating score | 30.98±8.1 | 35.76±7.13 | 33.15±8.03 | |
| SDS score | 44.93±9.81 | 46.78±8.85 | 45.77±9.43 | <.001 |
| SDS category | | | | |
| Depression | 691 (74.1) | 542 (69.4) | 1233 (71.9) | 0.032 |
| No depression | 242 (25.9) | 239 (30.6) | 481 (28.1) | |

SDS scores than men (p < 0.001). Notably, a slightly greater proportion of men (74.1%) were classified as depressed compared to women (69.4%), and this difference was statistically significant (p = 0.032). This finding suggests that while women had higher SDS scores overall, the prevalence of depression was slightly higher in men.

Table 2 displays the correlations among the research variables. The exercise score exhibited a significant negative correlation with BMI and body dissatisfaction. Emotional eating scores showed a significant positive correlation with body dissatisfaction and a negative correlation with exercise score. Depression was significantly negatively correlated with BMI and exercise scores, and positively correlated with emotional eating.

The findings presented in Table 3 indicate significant correlations (p < 0.05) between BMI and SDS scores, body dissatisfaction and emotional eating, as well as exercise score, and SDS score in each pairwise comparison—all of which demonstrated mediating effects.

The structural equation model depicted in Fig 1 elucidates the intricate interplay among BMI, body dissatisfaction, exercise, emotional eating behavior, and depression. The path diagram illustrates that exercise acted as a mediator between BMI and depression. Referring to Table 3, the mediating effect value of BMI on depression was -0.074, constituting 30% of the total effect (-0.247).

Additionally, exercise mediated the association between body dissatisfaction and emotional eating behaviors, with a mediating effect value of 0.074. This accounted for 19.8% of the total effect (0.373). Furthermore, emotional eating behavior mediated the link between exercise and depression, with a mediating effect of -0.004. This represented 8.5% of the total effect (-0.047).

Moreover, the model fitting index indicated that the structural equation model was satisfactory (AIC = 56057, BIC = 56155, P = 0.013). Table 4 provides examples of reporting results that mainly use fit indices. We used several fit indices to test the fit of the data model: (i) Fit index (CFI, AGFI, GFI, and TLI) >0.95 was considered satisfactory, and values between 0.90 and 0.95 were considered acceptable; (ii) Root mean square Error of Approximation (RMSEA), the range of

**Table 2. Two-by-two correlation of study variables.**

| Variables | 1 | 2 | 3 | 4 | 5 | 6 |
|---|---|---|---|---|---|---|
| 1. BMI | 1 | | | | | |
| 2. Body dissatisfaction | 0.787** | 1 | | | | |
| 3. Physical activity score | −0.136** | −0.241** | 1 | | | |
| 4. Emotional eating score | 0.037 | 0.096** | −0.058* | 1 | | |
| 5. External eating score | −0.044 | 0.064** | −0.053* | −0.222** | 1 | |
| 6. SDS score | −0.053* | 0.009 | −0.085** | 0.176** | −0.014 | 1 |

**P < 0.01;

*P < 0.05

**Table 3. Total and indirect effects among the studied variables.**

| Effect | Predictive variable | Result | β | std.β | SE | P |
|---|---|---|---|---|---|---|
| Total effect | BMI | SDS | −0.247 | −0.09 | 0.069 | 0.0003 |
| | Body dissatisfaction | EES | 0.373 | 0.107 | 0.086 | <0.0001 |
| | Physical activity score | SDS | −0.047 | −0.095 | 0.012 | <0.0001 |
| Indirect effect | BMI | SDS | −0.074 | −0.027 | 0.021 | 0.0005 |
| | Body dissatisfaction | EES | 0.074 | 0.021 | 0.036 | 0.039 |
| | Physical activity score | SDS | −0.004 | −0.009 | 0.002 | 0.0433 |

EES: Emotional eating score  SDS: Self-Depression Scale score  BMI: Body mass index

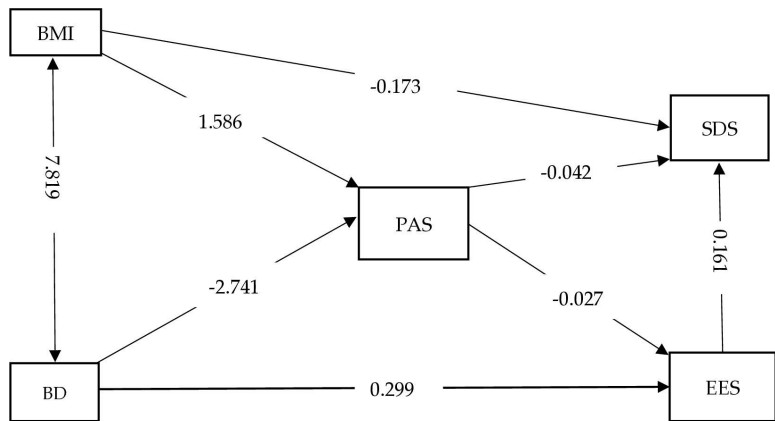

Note: BMI: Body mass index BD: Body dissatisfaction PAS: Physical activity score

EES: Emotional eating score SDS: Self-Depression Scale score

**Fig 1. Structural equation model of the relationship between BMI, body dissatisfaction, exercise, emotional eating behavior, and depression.**

**Table 4. Goodness of Fit Indices.**

|  | $\chi^2$ | df | GFI | AGFI | RMSEA | SRMR | CFI | TLI |
|---|---|---|---|---|---|---|---|---|
| Model | 8.632 | 2 | 0.998 | 0.985 | 0.043 | 0.009 | 0.996 | 0.982 |

AGFI = adjusted goodness of fit index; CFI = comparative normed fit index; df = degree of freedom; GFI = goodness-of-fit index; RMSEA = root mean squared error of approximation; SRMR = standardized root mean square residual; TLI = Tucker-Lewis index.

0.05–0.08 indicates a sufficient fit; Standardized root mean square residual (SRMR), values < 0.05 indicate a good fit and a range between 0.05 and 0.10 is acceptable [32].

## 4. Discussion

This study investigates the complex relationships between exercise, body dissatisfaction, emotional eating, and depression among Chinese college students using structural equation modeling. Our results demonstrate that exercise significantly mediates the relationship between BMI and depression, accounting for 30% of the total effect. Additionally, exercise mediates the relationship between body dissatisfaction and emotional eating, contributing to 19.8% of the total effect. Emotional eating behavior also partially mediates the relationship between exercise and depression.

To elucidate the intricate dynamics among body dissatisfaction, exercise, emotional eating, and depression, we employed structural equation modeling. We found that emotional eating partially mediated the relationship between physical activity and depression (Fig 1). Emotional eating behavior partially mediated the relationship between exercise and depression, accounting for 8.5% of the total effect. This similar to a study involving nearly 1,000 women at 22 universities in the New Orleans area, which revealed the significant role of emotional eating in linking BMI and depressive symptoms [33]. However, in our study, emotional eating did not mediate the relationship between depression and BMI, possibly due to the low obesity rate among college students in southern China, this contrasts with the high-risk obesity population in the referenced study [34]. Our mediation analysis revealed a significant change in the effect of exercise on depression when considering emotional eating, with emotional eating behaviors moderating the efficacy of exercise in

improving depression. This suggests that emotional eating behaviors may act as a moderating mechanism, explaining only some associations, while other factors remain unaddressed. Our findings highlight the potential psychological role of emotional eating behavior, particularly in regulating emotions and influencing depression risk. Moreover, the results indicate that emotional eating partially mediates the relationship between physical activity and depression. In this context, emotional eating scores may more accurately reflect failures in emotion regulation rather than actual changes in food intake [35]. Consequently, emotional eating is unlikely to result in actual changes in BMI. A Polish study where only 34% of participants ate healthy food when stressed, despite 80% reporting usually maintaining a healthy diet [36]. our findings are consistent with this observation and suggest that emotional eating behaviors serve as a mediating regulatory mechanism between exercise and depression but did not alleviate depression or mitigate the positive effects of exercise on depression.

Our model not only identified the mediating effect of emotional eating but also found that exercise plays an important role in the model. We found that physical exercise has the potential to mitigate depression risk (Table 2). A UK analysis of 15 prospective studies—including a systematic review involving nearly 200,000 adults—corroborated our results, revealing a negative dose-response relationship between physical activity and depression, with a steeper gradient as physical activity levels decreased [14]. Our model suggests that exercise's salutary impact on depression partially stems from its association with BMI, with higher-BMI individuals more inclined toward exercise. Notably, this contradicts prior studies indicating exercise reduces BMI [37]. In recent years, economic downturn, internet usage, and high-pressure educational methods have all contributed to people not having enough time for physical activities [38–40]. Coincidentally, the overall level of physical activity among our participants was considerably low, but some participants who were highly obese strengthened their physical activity, leading to the occurrence of this result. However, it is important to note that our study is cross-sectional, and we just draw conclusions about the temporality of the relationship between BMI and exercise. Reverse causality is possible, meaning that individuals with higher BMI might have recently increased their physical activity levels in an effort to change their body weight. Further longitudinal studies are therefore needed to establish the direction of causality.

In addition, exercise also has a mediating effect between body dissatisfaction and emotional eating, with those who are more dissatisfied with their bodies being more likely to engage in emotional eating behaviors. Specifically, higher levels of body dissatisfaction are associated with an increased likelihood of engaging in emotional eating behaviors. This finding suggests that body dissatisfaction may indirectly promote emotional eating by reducing self-esteem and self-efficacy, thereby diminishing the motivation to engage in physical activity. In this study, body dissatisfaction specifically refers to dissatisfaction with one's own body weight. When individuals are dissatisfied with their bodies, they may resort to emotional eating as a means of alleviating negative emotions, rather than opting for positive physical activities. This alternative behavior may stem from the immediate gratification provided by emotional eating, as well as the potential physical discomfort or psychological pressure associated with engaging in exercise [41,42]. This shows that exercise can regulate the conflicting relationship between body dissatisfaction and the tendency to engage in emotional eating behaviors. Therefore, exercise is crucial in regulating behavior and mental health.

The impact of emotional eating behaviors on depression warrants attention, as it correlates positively with depression risk (Table 2). A US review linked prior major depression experiences with heightened emotional eating [43]. Similarly, a study of over 1,300 Southern Chinese college students revealed that 52.7% reported high levels of emotion eating, and the prevalence of depressive symptoms was 18.6%. The study also demonstrated a strong correlation between emotional eating and depressive symptoms [44], which aligns with our findings. Emotional eating may serve as a stress coping mechanism, albeit transient and potentially exacerbating negative emotions [45]. Depression further exacerbates emotional eating behaviors [46]. Therefore, the mutual reinforcement between depression and emotional eating may be an important reason for the high levels of depression and emotional eating behavior in our results.

Moreover, women scored higher on emotional eating than men (Table 1). The transition to college life can be stressful for many women, leading them to resort to eating behaviors to relieve stress [41]. In addition, vulnerability of women individuals can also induce the occurrence of emotional eating behavior [47]. By contrast, men tend to relieve stress through exercise [48], rather than by eating. Moreover, Society holds higher expectations and stricter norms for women's body image and eating behaviors. Women are more frequently exposed to negative information about weight and appearance, which may lead to higher levels of emotional eating behaviors [49]. Therefore, it is crucial to pay attention to emotional eating behavior in women so as to prevent the development of eating disorders and depression.

We evaluated three eating behaviors, with emotional eating exhibiting the strongest positive correlation with body dissatisfaction (Table 2). As individuals near physical maturity and become increasingly concerned with body image, up to one-third of adolescents self-report body dissatisfaction [50]. This is worrisome because body dissatisfaction correlates with a heightened risk of emotional eating among college students. Furthermore, body dissatisfaction can exacerbate depressive moods. A study of college students in the northeastern US concluded that emotional eating, whether triggered by stress or depression, significantly correlates with body dissatisfaction [51]. This parallels our findings of elevated body dissatisfaction and emotional eating behaviors amid higher depression rates and overall lower physical activity levels among college students in southern China, which shows their ambivalence in actions and psychology. Women aspire to a slim figure, whereas men strive for a toned body. This gender disparity can be attributed to both sexes being influenced by prevailing aesthetic standards [52] that are shaped by various factors such as social media and ideals of beauty. One study confirmed the link between the pursuit of a slim body image and media consumption, with media serving as the primary promoter of this notion [53]. Recognizing the mediating role of emotional eating and the significant impact of body dissatisfaction can foster a deeper understanding of the bidirectional developmental relationship between exercise and depression, accelerating the development of more effective preventive and intervention strategies. Additionally, this insight can inform media campaigns aimed at improving aesthetic perceptions, alleviating body dissatisfaction, and preventing depression onset and progression.

On the whole, exercise significantly mediated the relationship between BMI and depression, accounting for 30% of the total effect. This suggests that physical activity has a substantial impact on reducing depression risk. Evidence indicates that trajectories of lean mass and fat mass remain stable during adolescence [54]. If higher BMI partly reflects higher fat mass, but lean mass represents a body shape preferred by society [24], this implies that exercise may improve overall mental health by enhancing self-esteem. Moreover, exercise also mediated the relationship between body dissatisfaction and emotional eating, contributing to 19.8% of the total effect. This indicates that engaging in regular physical activity can help mitigate the negative impact of body dissatisfaction on emotional eating behaviors. Emotional eating behavior partially mediated the relationship between exercise and depression, accounting for 8.5% of the total effect. This may be because 71.3% of the participants had low levels of physical activity, which weakened the association between emotional eating and depression. While this mediating effect is smaller compared to exercise, it still highlights the importance of addressing emotional eating behaviors in interventions aimed at reducing depression risk.

**Limitations of the study**

This study had several limitations. With 1,714 participants, the results may lack generalizability. Moreover, only Chinese college students were included, which restricts the applicability of these findings to other countries or age groups. Exercise intensity in the past month was subjectively determined by participation via a questionnaire [55], making it challenging to quantify the subsequent effect of these activities on depression. Additionally, our study is unable to control for confounding variables, such as socioeconomic status, nutrition, and environmental exposures. Therefore, our study has certain limitations in drawing conclusions. Our study employed a cross-sectional design, which limits our ability to draw conclusions about the temporality of the relationships between body dissatisfaction, exercise, emotional eating, and depression. Future longitudinal studies are needed to establish the direction of causality and to better understand the dynamic

interplay among these variables over time. Body dissatisfaction was measured using the difference between actual and ideal BMI. While this method provides a straightforward and quantifiable measure, it may not capture the full complexity of body dissatisfaction as assessed by more detailed psychological instruments. Future studies may consider using additional measures, such as self-report questionnaires or qualitative assessments, to provide a more comprehensive understanding of body dissatisfaction. While our findings suggest that addressing body dissatisfaction may be beneficial for improving mental health, our study did not test specific interventions. Future research should explore targeted interventions to reduce body dissatisfaction and emotional eating, and to promote physical activity among adolescents and young adults.

## 5. Conclusion

Depression, body dissatisfaction, and insufficient physical activity are prevalent challenges among college students in Southern China. Within our study, emotional eating behaviors, body image dissatisfaction, and physical inactivity were particularly notable among women. The findings presented in this paper underscore the influence of body dissatisfaction on physical inactivity and emotional eating behaviors among Chinese college students. Addressing body dissatisfaction and emotional eating holds significant promise for depression prevention. Moving forward, future research should involve selecting universities in representative cities by region to conduct surveys, aiming to comprehensively assess the overall depression status and body image of Chinese youth and propose constructive interventions.

## Supporting information

**S1 File. Data.**
(XLSX)

## Acknowledgments

We would like to express our appreciation to all the study participants.

During the investigation, investigators need to explain in detail to the research subjects that the data obtained from this investigation is only used for scientific research activities and will not disclose personal information. They should actively answer any unclear questions when students fill out the questionnaire and assist students in completing the questionnaire smoothly. All participants have signed informed consent forms.

## Author contributions

**Data curation:** Mingtao Chen, Renzhao Huang, Weiguo Chen, Huigen Liu, Ming Hao.

**Formal analysis:** Zhimin Yi.

**Funding acquisition:** Guoqiu Liu, Ming Hao.

**Writing – original draft:** Zhimin Yi.

**Writing – review & editing:** Ming Hao.

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
