## [Decision Letter · Decision Letter 0]

6 Feb 2025

PONE-D-24-33818Exploring Exercise, Emotional Eating, and Body Dissatisfaction in Depression RiskPLOS ONE

Dear Dr. Hao,

Thank you for submitting your manuscript to PLOS ONE. After careful consideration, we feel that it has merit but does not fully meet PLOS ONE’s publication criteria as it currently stands. Therefore, we invite you to submit a revised version of the manuscript that addresses the points raised during the review process.

We look forward to receiving your revised manuscript.

Kind regards,

Randy Wayne Bryner, Ed.D.

Academic Editor

PLOS ONE

**Journal Requirements:**

This study was supported by the Starting Research Fund from the Gannan Medical University�QD202121��Humanities and Social Sciences Fund from the Ministry of Education of China (22YJC630085), Humanities and Social Sciences of Jiangxi University in 2023�GL23114�.

3. In the online submission form, you indicated that The datasets used and analyzed during the current study are available from the corresponding author on reasonable request.

**Additional Editor Comments:**

Thank you for your submission. Comments have been received from external reviewers with particular expertise in the area of your manuscript. While we find the topic of your paper interesting, the reviewers have raised a number of concerns for the manuscript in its current form. Major revisions are required before the manuscript can be published in PLOS ONE.

We believe both reviewers have provided some valuable feedback that could be useful. As a result, you will see we are advising that you revise your manuscript based on the reviewers concerns. Please understand, the opportunity to provide a revised manuscript does not imply any guarantee that the revised work will yet receive a sufficient priority rating to be accepted for publication. That decision will ultimately be based on the comments and recommendations from the next set of reviews, should you decide to revise the manuscript.

Reviewers' comments:

Reviewer's Responses to Questions

**Comments to the Author**

1. Is the manuscript technically sound, and do the data support the conclusions?

Reviewer #1: Partly

Reviewer #2: Yes

2. Has the statistical analysis been performed appropriately and rigorously? 

Reviewer #1: Yes

Reviewer #2: I Don't Know

3. Have the authors made all data underlying the findings in their manuscript fully available?

Reviewer #1: No

Reviewer #2: No

4. Is the manuscript presented in an intelligible fashion and written in standard English?

Reviewer #1: Yes

Reviewer #2: Yes

5. Review Comments to the Author

**Reviewer #1:**  Dear authors,

Thank you for the opportunity to review your manuscript. I have a few recommendations for changes.

Generally, the manuscript is grammatically well written, but it does need to be proofed for errors and inconsistencies – for example, man instead of male, and a space after the last word and the bracket for citation.

The manuscript starts by highlighting the importance of depression among college students, but as you get into the moderators (PA, EES, body dissatisfaction), the justification for the overall study design seems to get a bit lost.

Overall, there’s a thorough discussion, but the relationships between these outcomes gets a little lost. You talk through each relationship individually, but there is not a cohesive explanation of all findings. A careful, thoughtful, well-worded explanation is needed that describes these mediating relationships, how it impacts depression, and then how the reader can use this information. Starting the manuscript, I would assume PA, EES, and body dissatisfaction mediate depression – upon ending the manuscript, it felt like you talked about each individual relationship, but now how they act together on depression.

Specific comments:

• Abstract:

o The abstract is lacking any data, please include specifics that are found in the results section.

o The first sentence of the abstract is incomplete.

• In the introduction:

o The sentence starting with “during college years” needs cited

o The sentence starting with “in addition to change sin physical activity” – whatever point you’re trying to make her is not fully explained.

• Methods

o “head portal sensitization” is an unfamiliar recruitment method, please full explain

o It says in a few places that participants did 3 questionnaires, but it looks like they did 4 – can you please clarify in the manuscript?

• Results and discussion:

o instead of “man” use “male”

o Mouth is the wrong word in the discussion

o The first paragraph needs to summarize what you found, then the second paragraph can dig into a discussion about one of the primary findings.

**Reviewer #2:**  In this study, Yi et al. investigated the relationship between BMI, body dissatisfaction, emotional eating, physical activity levels, and depression risk among Chinese college students. Whereas previous studies have investigated direct relationships between dietary behaviors, physical activity, body dissatisfaction and depression risk, the interrelationships of these variables have not been fully examined. They aimed to understand the interactive effects of body dissatisfaction and dietary and physical activity behaviors and their relative contribution to depression risk using structural equation modeling. They found that males had a slightly higher BMI and higher self-reported physical activity levels, whereas females had higher scores on all self-reported eating-related indices. Females had slightly higher self-reported depression scores, though a higher percentage of males were classified as having depression. BMI was positively correlated with body dissatisfaction and negatively correlated with depression scores. Physical activity was negatively correlated with BMI, body dissatisfaction, emotional eating, and depression scores. The relationships between BMI and depression and between body dissatisfaction and emotional eating were partially mediated by physical activity, and the relationship between physical activity and depression was partially mediated by emotional eating. They concluded that emotional eating and physical activity are important lifestyle-related behaviors to target to reduce depression risk in college students, and that body dissatisfaction influences both of these behaviors.

Key strengths of the manuscript are the use of structural equation modeling to examine the complex relationships between body dissatisfaction and lifestyle behaviors on depression risk, and the authors clearly and logically establish the rationale for the study and how it adds to the scientific literature.

Weaknesses of the manuscript include a lack of reporting about some details of the model and whether alternative models were considered, the need to justify why the model didn’t include a bidirectional relationship between emotional eating and depression, the validity of the body dissatisfaction and emotional eating scores for measuring intended behaviors, and the need for reorganization, elaboration, and clarification of some portions of the manuscript, as described in more detail below.

This manuscript is recommended for major revisions.

Major issues with the manuscript are as follows:

In the methods, the authors note that the model is “satisfactory” and list values related to model fit. The manuscript would be stronger with a table reporting more details about the model fit, as well as text explaining whether the authors considered alternative models and how they settled on this one. An invited review by Kang and Ahn published in Asian Nursing Research in 2021 presents checklists and guiding questions to address during the all stages of SEM-based studies to help improve reporting quality (Kang and Ahn, 2021). They recommend including tables showing goodness of fit indices and estimates of loadings and covariances of measurement errors.

The authors cite evidence in the introduction and discussion indicating a bi-directional relationship between emotional eating and depression (citations #19, #43). It’s unclear why the model does not include the potential for depression to increase emotional eating since there is reason to believe this relationship exists based on prior literature.

In this study the authors use the metric of a body dissatisfaction score, which is calculated as actual BMI minus desired BMI. Observed variables should be measured using tools that have been verified for accuracy and reliability (Kang and Ahn, 2021). It’s unclear if this metric is validated against other accepted measures of body dissatisfaction. The methods section includes a citation for Venckunas, 2016, in the description for collecting information about body dissatisfaction. However, that article does not measure body dissatisfaction or include the body dissatisfaction score metric; it’s a paper about trends in fitness and body size in Lithuanian children. A quick search of the literature for papers using this metric revealed only papers from the author’s lab and one other paper (Meyer, 2024) using that approach. Body dissatisfaction is related to more than someone’s desired weight, which as an indicator itself also doesn’t account for any body image disturbance related to body composition, weight distribution, muscular appearance or “tone,” or other physical attributes. Body dissatisfaction is a cognitive-affective psychological construct (Quittkat, 2019). The actual minus desired BMI metric doesn’t really seem to capture body dissatisfaction in this sense. The metric used by the authors is better defined as a weight discrepancy or BMI discrepancy. Indeed, the other paper that used actual minus desired BMI as an indicator of body dissatisfaction (Meyer, 2024) spent majority of the discussion talking about their findings in the context of discrepancy between current and desired BMI, rather than in the context of the feelings of being dissatisfied with one’s body.

The C-DBEQ and original DBEQ are validated instruments that have been used to quantify emotional eating behaviors in many research studies. However, in 2016, Bongers and Jansen published a review in Frontiers in Psychology that calls into question the validity of the DBEQ and other self-report questionnaires that measure emotional eating as being reflective of increased food intake in response to negative emotions (Bongers, 2016). They argue that evidence does not support a consistent link between emotional eating scores and increased food intake. Additionally, they point out that emotional eating can also occur in response to positive feelings in addition to negative feelings. Instead, they show that high emotional eating scores are related to high scores on several other eating-related scales, and thus may reflect other behaviors or traits, including lack of control, general eating concerns, a tendency to attribute overeating to negative affect, or learned cue reactivity. If interventions towards emotional eating are going to be effective, they must target actual behaviors or traits associated with the alterations in food intake and thus mood and depression risk and so the general concept of emotional eating may not be helpful. There is precedent for using the emotional eating scale used in this study, so a way that the authors could strengthen the manuscript would be to address this in the discussion of their findings. The authors should be cognizant of how well the construct used to measure emotional eating (body dissatisfaction, as well) actually represents the behavior or trait of interest when considering the relationships between emotional eating scores, dissatisfaction with body weight, physical activity levels, and depression risk and incorporate discussion of this potential limitation to the study.

The authors discuss how physical activity and emotional eating partially mediate the risk of depression, but there is not much reflection in the discussion about the explanatory power of each variable or the magnitude of associations observed in the study. The manuscript would be strengthened by translating these numbers into information more easily understood by those who are not experts in structural equation modeling in both the results and discussion sections.

The authors report data for muscle mass in kilograms in table 1 and report in the methods that muscle mass was assessed by questionnaire. It’s unclear why this measurement, which is arguably more difficult to know and self-report with accuracy than body weight, was not measured whereas body weight and height were measured by the authors. There are accessible tools available for measuring body composition with reasonable accuracy. For example, the Omron HBF-516 home body composition scale has good accuracy when validated against criterion methods (Siedler et al, 2023) and is commercially available and inexpensive. Self-reported muscle mass is unreliable and doesn’t add helpful information to the study.

Minor issues, separated by section of manuscript:

Abstract: Includes grammatical errors and incomplete sentences, requires review

Introduction

o At the end of the first paragraph, the authors state, “However, there are few studies on the combined effects of exercise and dietary behaviors on depression.” The authors should support this claim by including citations for these studies.

o Paragraph 2, near middle: The authors write, “Research indicates that exercise significantly impacts the severity and relief of depression [14, 15].” Citation 15 appears to be irrelevant to this claim, as the topic of the paper appears to be about immunity.

o Paragraph 3: The authors write in sentence two, “In 2022, according to a study in eating disorders has risen by 80% over the past 5 years [18].” This sentence is incomplete and appears to be missing a word.

o Paragraph 4: The authors write, “College students body dissatisfaction is closely related to eating behaviors and exercise [24, 25].” Does reference 25 demonstrate this point? Also, very minor, but in this context the word students is possessive and should end with an apostrophe (students’).

Methods

o Sentence 3: “In accordance with these policies, investigators provide research participants…” Provide should be provided.

o Paragraph 2: What’s head portal sensitization?

o Paragraph 3, last sentence: the word “date” should be “data”

o C-DEBQ: What’s the range of scores possible on this instrument and are there criterion thresholds for determining what’s a “high” score vs a “low” score? It’s hard to interpret the scores presented in table 1 without context about the range of scores possible.

o Directly after the C-DEBQ description: The sentence stating “A questionnaire was utilized….monthly living expenses, and other relevant details” should be moved to the start of the methods with the other general information assessed by questionnaire. Also, the phrase “and other relevant details” is vague and not repeatable. The authors should include specific information for all details assessed on the questionnaire, or provide justification for why the specific nature of the additional information is not listed.

o Section 2.5, Data Collection: The authors note that they removed invalid questionnaires. What’s the basis for determination of invalidity?

Results

o Paragraph 1, sentence 2 begins “Man” and should read “Men.” This paragraph ends with information about the mean SDS scores being higher in women than men. However, it’s notable that, according to the data in table 1, a slightly greater proportion of males (74.1%) were classified as depressed than were females (69.4%) and this difference was statistically significant (p=0.032) but this was not noted in the text of the results.

o There appear to be typos in table 1 where data are presented as n (%). The numbers presented do not equal the percentage of the total sample indicated in the parentheses for values in the data for BMI category, activity level category, and SDS category.

o Table 2 does not report information about measurement error.

o As described earlier in the major issues section, the results section would be stronger if it included data tables with information about the model fit and measurement model, such as that presented in Kang and Ahn, 2021, tables 3 and 4.

Discussion

o The manuscript would be strengthened by a sentence or two at the start of the discussion summarizing the key findings of this paper as they relate to the specific aim and hypothesis of this study and what it adds to the literature before going into deeper discussion of the study.

o In paragraph 2, the authors discuss how exercise plays a role in the model. The second sentence notes the “potential of positive exercise in mitigating depression risk.” It’s unclear what the authors mean by “positive” exercise. Please clarify or reword. Additionally, in sentences 4 and 5 of this paragraph the authors state that their results contradict prior studies indicating that exercise reduces BMI. However, the authors conducted cross sectional study of participants, no conclusions can be drawn about the temporality of the relationship between BMI and exercise since reverse causality is possible, i.e. higher BMI individuals recently started exercising more frequently in effort to change body weight. It seems that this may be what the authors are suggesting in sentence 7, but it could be reworded for clarity.

o Paragraph 3, sentence 3: This sentence is confusing and appears to be missing some words and/or punctuation, needs revised for clarity.

o In paragraph 4, the authors argue that the transition to college causes changes in eating behaviors and that women and men cope differently and cite three studies for support (#44, 45, 46). This makes me wonder who are the populations studied in these papers, and are students in China expected to respond in similar ways, or could cultural differences result in other coping behaviors that may be different? Sociocultural factors have strong influences on eating behaviors and practices.

o Paragraph 5: The authors write near the middle of the paragraph, “Perhaps a relatively large proportion of our survey respondents experienced body dissatisfaction that then triggered a qualitative change in the results. Second, unlike most other research methods, the current research methods employed in the actual and ideal BMI difference for representing body dissatisfaction may also lead to identifying different causes.” It’s unclear what the authors mean by triggered a qualitative change in the results. Additionally, related to earlier concerns about the validity of the body dissatisfaction metric used in this study, there is no justification in the methods for why the authors chose their metric for body dissatisfaction instead of a method used in prior studies on body dissatisfaction, and the discussion of this potential limitation is restricted to the two sentences quoted above and the subsequent two sentences in the discussion. Elaboration would strengthen the manuscript.

o Limitations of the study: Elaboration in this section, or integrating an expanded discussion of limitations within the other text of the discussion, would strengthen the manuscript. Also, there is a typo in sentence 3; the word mouth should be month.

My References

•Bongers P and Jansen A. Emotional Eating is Not What You Think It Is and Emotional Eating Scales Do Not Measure What You Think They Measure. Front. Psychol., 07 Dec 2016.

•Kang H and J-W Ahn. Model Setting and Interpretation of Results in Research Using Structural Equation Modeling: A Checklist with Guiding Questions for Reporting. Asian Nursing Research. 2021;15:157-162

•Meyer E, Lönnroth K, Forsell Y, Lagerros YT. Discrepancy between Current, Desired, and Ideal Body Mass Index in Persons with Obesity: A Swedish Population-Based Study. Obes Facts. 2024;17(1):72-80. doi: 10.1159/000535198. Epub 2023 Nov 20. PMID: 37984350; PMCID: PMC10836935.

•Quittkat HL, Hartmann AS, Düsing R, Buhlmann U, Vocks S. Body Dissatisfaction, Importance of Appearance, and Body Appreciation in Men and Women Over the Lifespan. Front Psychiatry. 2019 Dec 17;10:864. doi: 10.3389/fpsyt.2019.00864. PMID: 31920737; PMCID: PMC6928134.

•Siedler MR, Rodriguez C, Stratton MT, Harty PS, Keith DS, Green JJ, Boykin JR, White SJ, Williams AD, DeHaven B, Tinsley GM. Assessing the reliability and cross-sectional and longitudinal validity of fifteen bioelectrical impedance analysis devices. Br J Nutr. 2023 Sep 14;130(5):827-840. doi: 10.1017/S0007114522003749. Epub 2022 Nov 21. PMID: 36404739; PMCID: PMC10404482.

6. PLOS authors have the option to publish the peer review history of their article (what does this mean? ). If published, this will include your full peer review and any attached files.

**Do you want your identity to be public for this peer review?** For information about this choice, including consent withdrawal, please see our Privacy Policy .

Reviewer #1: No

Reviewer #2: **Yes: ** Myra Woodworth-Hobbs

---

## [Author Response · Author response to Decision Letter 1]

4 Mar 2025

I have uploaded the file of response to the reviewer.

---

## [Decision Letter · Decision Letter 1]

25 Mar 2025

Exploring Exercise, Emotional Eating, and Body Dissatisfaction in Depression Risk :using structural equation modeling

PONE-D-24-33818R1

Dear Dr. Hao,

We’re pleased to inform you that your manuscript has been judged scientifically suitable for publication and will be formally accepted for publication once it meets all outstanding technical requirements.

Kind regards,

Randy Wayne Bryner, Ed.D.

Academic Editor

PLOS ONE

Additional Editor Comments (optional):

Thank you for your careful review and addressing the comments of the reviewers.

Reviewers' comments:

Reviewer's Responses to Questions

**Comments to the Author**

1. If the authors have adequately addressed your comments raised in a previous round of review and you feel that this manuscript is now acceptable for publication, you may indicate that here to bypass the “Comments to the Author” section, enter your conflict of interest statement in the “Confidential to Editor” section, and submit your "Accept" recommendation.

Reviewer #1: All comments have been addressed

2. Is the manuscript technically sound, and do the data support the conclusions?

Reviewer #1: Yes

3. Has the statistical analysis been performed appropriately and rigorously? 

Reviewer #1: Yes

4. Have the authors made all data underlying the findings in their manuscript fully available?

Reviewer #1: Yes

5. Is the manuscript presented in an intelligible fashion and written in standard English?

Reviewer #1: Yes

6. Review Comments to the Author

Reviewer #1: Thank you for addressing reviewers comments. I appreciate your thorough revisions and have no further concerns.

7. PLOS authors have the option to publish the peer review history of their article (what does this mean? ). If published, this will include your full peer review and any attached files.

**Do you want your identity to be public for this peer review?** For information about this choice, including consent withdrawal, please see our Privacy Policy .

Reviewer #1: No

---

## [Editor Report · Acceptance letter]

PONE-D-24-33818R1

PLOS ONE

Dear Dr. Hao,

I'm pleased to inform you that your manuscript has been deemed suitable for publication in PLOS ONE. Congratulations! Your manuscript is now being handed over to our production team.

Kind regards,

on behalf of

Dr. Randy Wayne Bryner

Academic Editor

PLOS ONE